# Apoptosis and (in) Pain—Potential Clinical Implications

**DOI:** 10.3390/biomedicines10061255

**Published:** 2022-05-27

**Authors:** Hugo Ribeiro, Ana Bela Sarmento-Ribeiro, José Paulo Andrade, Marília Dourado

**Affiliations:** 1Community Support Team in Palliative Care–Group of Health Centers Gaia, 4430-043 Vila Nova de Gaia, Portugal; haribeiro@arsnorte.min-saude.pt; 2Coimbra Institute for Clinical and Biomedical Research (iCBR)-Group of Environment Genetics and Oncobiology (CIMAGO), FMUC, 3000-548 Coimbra, Portugal; 3Centro de Estudos e Desenvolvimento dos Cuidados Continuados e Paliativos da Faculdade de Medicina da Universidade de Coimbra, 3000-548 Coimbra, Portugal; 4Programa Doutoral em Cuidados Paliativos da Faculdade de Medicina da Universidade do Porto, 4200-319 Porto, Portugal; 5Laboratory of Oncobiology and Hematology (LOH) and University Clinic of Hematology, Faculty of Medicine, University of Coimbra, FMUC, 3000-548 Coimbra, Portugal; 6Center for Innovative Biomedicine and Biotechnology (CIBB), 3000-548 Coimbra, Portugal; 7Centro Hospitalar e Universitário de Coimbra (CHUC), 3004-561 Coimbra, Portugal; 8Department of Biomedicine–Unity of Anatomy, Faculty of Medicine of Universidade of Porto, 4200-319 Porto, Portugal; jandrade@med.up.pt; 9CINTESIS@RISE, Faculty of Medicine, University of Porto, 4200-319 Porto, Portugal

**Keywords:** apoptosis, pain, cell signaling, oxidative stress, inflammation, biomarkers, targeted therapies, transient receptor potential (TRP) cation channels

## Abstract

The deregulation of apoptosis is involved in the development of several pathologies, and recent evidence suggests that apoptosis may be involved in chronic pain, namely in neuropathic pain. Neuropathic pain is a chronic pain state caused by primary damage or dysfunction of the nervous system; however, the details of the molecular mechanisms have not yet been fully elucidated. Recently, it was found that nerve endings contain transient receptor potential (TRP) channels that sense and detect signals released by injured tissues and respond to these damage signals. TRP channels are similar to the voltage-gated potassium channels or nucleotide-gated channels that participate in calcium and magnesium homeostasis. TRP channels allowing calcium to penetrate into nerve terminals can activate apoptosis, leading to nerve terminal destruction. Further, some TRPs are activated by acid and reactive oxygen species (ROS). ROS are mainly produced in the mitochondrial respiratory chain, and an increase in ROS production and/or a decrease in the antioxidant network may induce oxidative stress (OS). Depending on the OS levels, they can promote cellular proliferation and/or cell degeneration or death. Previous studies have indicated that proinflammatory cytokines, such as tumor necrosis factor-α (TNF-α), play an important role in the peripheral mediation of neuropathic pain. This article aims to perform a review of the involvement of apoptosis in pain, particularly the role of OS and neuroinflammation, and the clinical relevance of this knowledge. The potential discovery of new biomarkers and therapeutic targets can result in the development of more effective and targeted drugs to treat chronic pain, namely neuropathic pain. Highlights: Oxidative stress and neuroinflammation can activate cell signaling pathways that can lead to nerve terminal destruction by apoptosis. These could constitute potential new pain biomarkers and targets for therapy in neuropathic pain.

## 1. Introduction

Almost 20% of the European population suffers from chronic or intermittent pain, resulting in patient suffering and disability, and increases the economic burden on society [1]. In this context, severe pain is a major health care problem.

Neuropathic pain is a condition caused by a lesion or disease of the somatosensory system in agreement with the International Association for the Study of Pain (IASP) [2]. The neuropathic pain can be peripheral following peripheral nerve injury, postherpetic neuralgia, trigeminal neuralgia, painful radiculopathy, or painful polyneuropathy caused by metabolic and infectious diseases or chemotherapy. It can be central if associated with a spinal cord or encephalic lesion [3]. This type of chronic pain has a prevalence in the population between 7 to 10% [4,5]. Contrary to nociceptive pain, common analgesics are frequently unable to alleviate neuropathic pain. It is accompanied by hyperalgesia, allodynia, spontaneous pain, and causalgia [6], which can persist for years [7]. Therefore, neuropathic pain can become a chronic condition and hardly bearable. In extreme cases, it may lead to depression, anxiety, sleep disturbances, and impaired cognition [8]. 

Numerous mechanisms are involved in the pathogenesis of neuropathic pain, such as peripheral and central sensitization, changes in transient receptor potential (TRP) cation channels, ectopic activity in primary afferent fibers, and disturbance of the excitatory and inhibitory signaling [9] However, there is increasing evidence that neuroinflammation [10] and oxidative stress related to mitochondrial dysfunction and apoptosis are critically involved in inducing and maintaining neuropathic pain [9,11]. Apoptotic programmed cell death is now thought to be a principal pathway of neuronal death in many neurodegenerative diseases, including Alzheimer’s, Parkinson’s, and Huntington’s diseases, and also in neuropathic pain [12]. Mitochondria have a central role in apoptosis [12] as the permeabilization of the mitochondrial membranes leads to the release of cytochrome c and of the complex formation that begins the cleavage and activation of proteases known as caspases [13], inactivating molecules essential for cell survival. Other molecules that mediate a program of cell suicide are activated [13]. The reduction of the membrane potential of mitochondria, the generation of free radicals and reactive oxygen species (ROS), and intracellular acidification are some other features of apoptosis [12]. Supporting the idea that apoptosis occurring in sensory neurons may contribute to neuropathic pain conditions, multiple apoptosis-related genes were observed to be upregulated in the dorsal root ganglion in a spinal nerve ligation model of neuropathic experimental pain [14].

Our aim with the present narrative review is to understand the fine molecular mechanisms involving apoptosis and neuropathic pain and the relationship to nociceptive signaling, oxidative stress, and neuroinflammation. Special attention is paid to the clinical relevance of biomarkers and the development of potential new therapeutic targets. 

## 2. Definition and Mechanisms of Pain—A Brief Review

The IASP conducted a review about the definition of pain in 2020, and it is now considered as an “unpleasant sensory and emotional experience associated with, or resembling that associated with, actual or potential tissue damage” [15].

It was Charles Sherrington, winner of the Nobel Prize in Medicine in 1932, who divided for the first time the mechanisms of propioception, exteroception, and interoception [16], with pain, temperature, and pressure also part of exteroception [17].

Nociception refers to the processing of a noxious stimulus resulting in the perception of pain by the brain. Various neurotransmitters, nociceptors, and cellular components are involved in the mechanisms of transduction, perception, transmission, and modulation of pain [18], as shown in Figure 1. 

Transduction is the conversion of a noxious stimulus (mechanical, chemical, or thermal) into electrical energy by a peripheral nociceptor (free afferent nerve ending). Transmission describes the propagation through the peripheral nervous system via first-order neurons. Modulation occurs when first-order neurons synapse with second-order neurons in the dorsal horn cells of the spinal cord, and perception is the cerebral cortical response to nociceptive signals that are projected by third-order neurons to the brain [19].

Pain depends on a multifactorial system with multiple pathways for numerous cortex areas, the cortical body matrix [20], which explains why it is a multidimensional experience that can significantly impair an individual’s quality of life [21]. 

The most abundant pain sensors are in the skin and are called the transient receptor potential cation channels (TRP) [22]. The TRP family contains many members, such as the vanilloid receptors (TRPV), the canonical receptors (TRPC), the ankyrin repeat receptors (TRPA), the polycystin recaptors (TRPP), the melastatin receptors (TRPM), and the mucolipin receptors (TRPML) [22]. TRP channels are activated by histamine, bradykinin, fatty acid metabolites, endocannabinoids, cannabinoids, adenosine triphosphate (ATP), acid, heat, cold, prostaglandins, cytokines, capsaicin, monoterpenoids, among other stimuli. The initial sensation can be heat, cold, pain, or itching that can be several minutes in duration. Usually, prolonged activation of the TRP channels (for more than 20–30 min or so) results in long-term deactivation and pain relief [22]. Recently, nine TRP channels (TRPA1, TRPC5, TRPM2, TRPM4, TRPM7, TRPV1, TRPV2, TRPV3, and TRPV4) have been demonstrated to be activated by oxidative stress [23] and in neurons related to nociception (as in dorsal root ganglion (DRG) and trigeminal ganglia neurons), the expression levels of four TRP channels (TRPA1, TRPM2, TRPV1, and TRPV4) are high [11]. From [24,25], TRPV1 and TRPA1 have been demonstrated in neuropathic pain associated with diabetes or the administration of chemotherapeutics, probably mediated by the synthesis of reactive oxygen and nitrogen species [26,27], which are well-known TRPA1 activators [28,29]. Further, TRP channels have been associated with pain in neurodegenerative disorders (NDD) through oxidative stress-induced cell death, particularly related to inflammation and calcium homeostasis deregulation [29]. All these processes are connected with microglial activation, a proposed mechanism inductor of inflammation and neuropathic pain [30], which leads to neuronal degeneration and cell death in NDD, and simultaneously to neuropathic and inflammatory pain [31,32].

Mitochondria play important roles in a myriad of cellular processes, and mitochondrial dysfunction has been implicated in multiple neurological disorders [7]. The five major mitochondrial functions (the mitochondrial energy generating system, ROS generation, mitochondrial permeability transition pore, apoptotic pathways, and intracellular calcium mobilization) may play critical roles in neuropathic and inflammatory pain [7]. 

Oxidative stress is a central mediator of apoptosis, neuroinflammation, metabolic disturbances, and bioenergetic failure in neurons [33]. Oxidative stress and apoptosis, as observed in the DRG neurons, play an essential role in inducing and developing pain hypersensitivity [34]. 

It should also be noted that apoptosis is also regulated by proinflammatory cytokines, such as Interferon gamma (IFN-g), tumor necrosis factor-α (TNF-α), interleukina-1 beta (IL-1 b), and IL-6 [35]. The roles of IL-1 and TNF- α in apoptosis in neurodegenerative diseases such as Alzheimer’s disease were previously described [35,36]. More importantly, it is known that these proinflammatory cytokines have crucial roles in the modulation of neuropathic pain [35,37].

## 3. Apoptosis Pathways as Mediators of Pain Formation

### 3.1. Apoptosis Cell Signaling Pathways

The homeostasis of a multicellular organism depends on cell turnover, as it is related to the capacity to maintain the equilibrium between cell proliferation and death [38,39].

Physiological cell death occurs primarily through an evolutionary and conserved form of cellular suicide, called apoptosis (from the Greek word for decay, as with leaf fall, for example), in response to a series of intrinsic and extrinsic stimuli [38,39]. It was discovered in 1952. However, in 1972, the Australian pathologist John FR Kerr and the Scots Andrew H. Wylie and Alistair R. Currie showed the importance of this cell death process in developing an adult organism [38]. Unlike the other type of death, cell necrosis, in which the cell is a passive victim, cell apoptosis is an active form of death, in which the cell expends energy to carry out this genetically programmed cellular suicide process. For this reason, it is often called programmed cell death [38,40].

The induction of apoptosis can be divided into three steps: the inducing agent’s interaction with the cell, the biochemical transduction of the death signal, and the execution by the apoptotic machinery. Different extracellular signals, such as an increase in ROS production [41] and/or a decrease in antioxidants resulting in oxidative stress, inflammatory mediators such as TNF- α and IL1-β [42], moderate hyperactivity stimulation of the N-methyl-D-aspartate (NMDA) receptors [1], and the endoplasmic reticulum stress [43] could activate signal transduction pathways that converge on a final common pathway leading to the execution phase of cellular apoptosis. At least two pathways are involved, the mitochondria-dependent pathway (intrinsic or mitochondrial pathway), which is mediated by mitochondria and the B-cell lymphoma-2 (BCL-2) family proteins, and the extrinsic or membrane pathway involving external signaling via membrane receptors from the tumor necrosis factor (TNF) family, such as TNF-R1, FAS, TNF-apoptotic inducing ligand receptors 1 and 2 (TRAIL-R1 and 2, or death receptors DR4 and 5 [44], and the low-affinity nerve growth factor (NGF) receptor, p75-NTR [27,45]. Moreover, the p38MAPK pathway causes the activation of apoptosis in neuronal cells, inducing pain (such as bone cancer pain-related hyperalgesia) [46].

However, all the pathways culminate in the activation of a series of cysteine proteases, called caspases. The effector caspases, such as caspase 3, cleave a wide variety of protein targets that are responsible for the characteristic changes in apoptotic cells. These include cell shrinkage, nuclear condensation, membrane blebbing, fragmentation into apoptotic bodies, and membrane changes that can lead to phagocytosis of the affected cell [25,33]. An alternative, proinflammatory form of apoptosis [40] in macrophages infected with Salmonella or Shigella, termed “pyroptosis” [47]. Since its discovery, pyroptosis has been observed in the central nervous system [48] and the cardiovascular system, suggesting that this form of cell death is biologically significant. [46,47].

The knowledge of this active physiologic mode of cell death has profound implications on our understanding of several diseases. In fact, the deregulation of apoptosis is involved in the development of several diseases, such as cancer [49], neurodegenerative diseases as Alzheimer’s and Parkinson’s, and autoimmune or rheumatoid arthritis. Recent evidence suggests that apoptosis may be involved in pain, namely neuropathic pain [1].

The involvement of apoptosis in pain and the clinical relevance of this knowledge in the discovery of new potential biomarkers and therapeutic targets may be useful for the development of more effective and targeted drugs to treat pain.

#### 3.1.1. The Extrinsic or Membrane Pathway of Apoptosis

The activation of membrane receptors of the TNF family, namely TNF-R1, FAS, TRAIL-R1/R2 (or death receptor 1 and 2, DR1/2) [44] and NGF-R/p75-NTR [45] by the respective ligands, TNF-alpha, FAS-L, TRAIL, and NGF, triggers the activation of enzymes called caspases, which are the effectors of the apoptotic process (Figure 2). 

These cell surface receptors are mainly type I transmembrane proteins (TRAIL-DR4 and DR5 are the exception, as they are type II), which have cysteine residues in their extracellular domain. They are activated by oligomerization upon binding of the respective ligand to their extracellular domain. The cytoplasmic domain of these TNF family receptors shares a sequence with about 60 to 70 amino acid residues, which has been called the death domain. This death domain interacts with other proteins possessing the same domain, namely with FADD (FAS-associated death domain) proteins, in the case of FAS, or TRADD (TNF receptor-associated death domain), in the case of TNF-R1. For apoptosis to occur, an N-terminal death domain effector is required, through which these proteins bind to identical amino acid sequences in other proteins, recently identified as procaspase 8 that is cleaved with the consequent release of the active caspase 8 (Figure 2). Subsequently, caspase 8 activates caspase 1, formerly called ICE (interleukin converting enzyme). Caspase 1, in turn, activates caspase 3 by cleaving the respective procaspase, which is the true executor of apoptosis (Figure 2). The activation of caspase 3 leads to the inactivation of a factor that inhibits deoxyribonucleic acid (DNA) fragmentation, leading to its cleavage by the endonuclease responsible for the internucleosomal cleavage of chromatin, characteristic of cells undergoing apoptosis. 

The low-affinity p75 neurotrophin receptor (NGF-R/p75-NTR) also activates the extrinsic or membrane pathway of apoptosis through its ligand, the nerve growth factor (NGF). NGF is one of the neurotrophins (NTs) family members, discovered in 1950, and is essential for the neuronal survival, growth, and development of neurons as well as the maintenance of neuronal phenotype in the mature nervous system via interaction with specific nerve surface receptors, such as tyrosine receptor kinases A (TRK-A). Although NGF is a well-known neurotrophic factor, it also acts as a mediator of pain (namely in orofacial pain), itching, and inflammation by inducing cell death through the low-affinity p75 neurotrophin receptor (NGF-R/p75-NTR), which activates caspase 8 (Figure 2) [45].

#### 3.1.2. The Intrinsic or Mitochondrial Apoptotic Pathway

In view of the above, caspase 8 appears to be the first direct signaling link between events at the plasma membrane level and the apoptotic execution machinery. Alternatively, caspase 8 can act on the so-called mitochondrial transient permeability pore (MTPP) through BIDt, linking mitochondria to apoptotic execution. This binding was initially suspected after the existence of a protein in the inner/outer membrane of the mitochondria with apoptotic inhibitory functions, the BCL-2 protein, was demonstrated [50,51]. Other members of this family of proteins with antiapoptotic functions include the BCL-XL, BCL-W, MCL-1, and A1 proteins, among others. Contrary to the action of these proteins, there are others with proapoptotic functions, of which we highlight the BAX, BAD, BID, BIK, and BAK proteins, also members of this same family, but for which increased expression leads to cell death. Although the BCL-2 protein appears to be exclusively membrane-bound, particularly in the mitochondria, other related proteins, such as the BID and BAD proteins, are cytoplasmic, and are transported to the mitochondria during the process of apoptosis [52]. The change in mitochondrial membrane potential with the opening of the MTPP leads to the release of cytochrome c, a mitochondrial inner membrane protein, into the cytoplasm, which induces cell death by apoptosis. Cytochrome c binds to apoptotic protease activating factor (APAF-1), leading to a conformational change in the latter and forming a protein complex with activated caspase 9, called the apoptosome. Caspase 9 cleaves and activates other effector caspases, namely caspase 3, as mentioned above, triggering cellular apoptosis. In turn, caspases are inhibited by the inhibitor of apoptosis proteins (IAPs), such as survivin and XIAP (X-linked IAP), which are modulated by the proapoptotic proteins, SMAC (second mitochondria-derived activator of caspase) and DIABLO (direct inhibitor of apoptosis-binding protein with low PI) (Figure 1) [53,54,55]. Thus, the susceptibility of a cell to enter into apoptosis is determined, in part, by the relative concentrations of the pro and antiapoptotic proteins, namely the BCL-2 family members.

Death pathways may also be triggered by excessive stimulation of the NMDA receptors that may result in necrosis (strong receptor hyperactivity) or apoptosis (moderate receptor hyperactivity), making NMDA receptors potential therapeutic targets in pain by using antagonists of these receptors to reduce excitotoxic neuronal death [1]. Moreover, p73, a pivotal gene in DNA damage belonging to the p53 family, regulates neuronal cell differentiation and oligodendrocyte precursors. This gene regulates the transcription of two protein isoforms with opposite effects on the regulation of apoptosis: TAp73, with proapoptotic function, and DeltaNp73, with antiapoptotic activity. The balance between the two protein isoforms and, thus, the control of cell proliferation and death are affected by the ubiquitin–proteasome degradation system and mediated by ligases of NEDD family [1]. 

As previously mentioned, oxidative stress (OS)/ROS, generated exogenously or endogenously, are involved in neuropathic pain (NP) through the intrinsic and extrinsic apoptotic pathways. Further, TRPM2 and TRPM7 channels can induced cell death mediated by oxidative stress (which could be also be related to inflammation), contributing to the observed relationship between pain, neurodegeneration, and TRPs [29,46,56].

### 3.2. Ubiquitin Proteasome Pathway and Apoptosis

Regulatory proteins involved in normal cell proliferation and differentiation are degraded by a proteolytic pathway, which involves lysosomal enzymes, and a non-lysosomal pathway, known as the ubiquitin–proteasome pathway (UPP) [56]. 

The UPP encompasses several enzymes, namely the ubiquitin-activating enzyme (E1), the ubiquitin-conjugating enzyme (E2), and a ligase (E3) that catalyzes the binding of ubiquitin to the target protein, the deubiquitinating enzyme (DUB), and proteasomes (namely the 26S proteasome) [57]. 

The degradation of proteins by the UPP involves two successive and distinct steps: first, several ubiquitin molecules bind to the target protein to be subsequently degraded in the proteasomes. Some of the regulatory proteins with levels are controlled by this degradative pathway include proteins involved in cell death by apoptosis, namely the BCL2 family proteins such as BAX protein [56], and the C-terminal fragment of the BID protein the BIDt [58], and the IAP family and the inhibitor of the NF-kB, IkB [59,60]. 

The UPP is relevant in neural development and brain structure and also in maintenance of their functions, and is implicated in synaptic plasticity, formation, and maintenance of memory. Recently, it was found that the dysfunction of this proteasomal degradation pathway leads to changes in cell death by apoptosis in diverse cell types and may be implicated in the pathogenesis of several diseases, either through hypofunction, as in neurodegenerative diseases [61,62], or by hyperfunction, as in the case of neoplastic diseases [57,62]. More recently, ubiquitin system dysfunction has been implicated in chronic pain, mainly in neuropathic pain (NP) and inflammatory pain, through ubiquitination modified protein receptors and ion channels to affect synaptic activity and efficiency [57]. 

One of the most important enzymes is the ubiquitin-conjugating enzyme E2B (Ube2b), also known as RAD6B. Ube2b is a member of the ubiquitin-conjugating enzyme family that has been identified to be vital in neural DNA double-strand DNA breaks (DSBs). Further, the Ube2b modulates the DNMT3a (DNA methyltransferase 3 alpha) ubiquitination degradation and *CaMKK1* (calcium/calmodulin-dependent protein kinase) gene promoter demethylation increasing CaMKK1 level. Moreover, the depletion in Ube2b reduces H2B (histones H2B) ubiquitination, contributing to elevation of the instability of the genome and neurodegeneration [62]. These studies indicate that Ube2b is essential in modulating nerve function, being a potential target to treat neuropathic pain. However, the role of Ube2b in neuropathic pain is not yet clarified [63]. Other UPP enzymes are the E3-ubiquitin protein ligases, a group of proteins that mediate ubiquitination and regulate chronic inflammatory pain via controlling the level of substrate proteins and then possibly adjust synaptic efficacy. The roles of these enzymes are being studied in inflammatory pain. The deubiquitin enzymes (DUBs) are ubiquitin-specific proteases that remove the ubiquitin groups from the degraded target proteins, improving protein stability. One of these DUBs is USP5, which increases the Cav3.2 protein level and calcium current contributing to inflammatory pain. The knockout of UPS5 by shRNA can increase Cav3.2 ubiquitination, decrease Cav3.2 protein level, and decrease calcium current. Thus, targeting Cav3.2 deubiquitination by USP5 is a potential therapeutic target for pain associated with inflammatory disorders [62].

Moreover, the nuclear factor-κB (NF-κB) is a key regulator of molecules and pathways important for inflammation and pain [64,65,66]. NF-κB can exist as homo or heterodimers composed of the Rel family proteins (the p65/p50 subunits in humans) and is bound to the κB inhibitor (IκBα) in the cytoplasm, when in an inactive state. Upon activation by proinflammatory cytokines and growth factors, among others, the IκB kinase (IKK) phosphorylates IκBα, targeting its degradation in the UPP. As a result, IκBα releases the p65/p50 complex, which subsequently translocates to the nucleus and initiates the transcription of several genes, including inflammatory genes such as TNFα, IL-1β, and cyclooxogenase-2 (COX-2) that directly or indirectly influence pain [64,65,66]. Thus, targeting the NF-kB pathway, namely by inhibiting the IkB degradation in the UPP could be a new therapeutic approach to treating pain [43,64,65].

### 3.3. Endoplasmic Reticulum Stress Signalling and Apoptosis

Endoplasmic reticulum (ER) stress causes the activation of caspase signaling pathway-dependent apoptosis in neuronal cells and induces bone cancer pain-related hyperalgesia.

The ER is the cellular organelle in which protein folding, lipid biosynthesis, and calcium storage take place. Many factors can cause an imbalance in the homeostasis of ER function, resulting in ER stress. Such ER stress initiates an evolutionarily conserved signaling cascade called the unfolded protein response (UPR), which is a self-protective signaling pathway. The accumulation of unfolded proteins in the ER can activate the UPR to restore ER function. Besides the increase in ER stress as a self-protective signal transcription pathway after mild injury, the failure of this system to relieve prolonged or excessive ER stress may cause cell apoptosis, and plays an important role in neuropathic pain, specifically in rat models of bone cancer pain [43].

### 3.4. Oxidative Stress

When the synthesis of oxidative species is superior to the capacity of the cells to counteract them, there is the occurrence of oxidative stress. The generation of reactive oxygen species (ROS) in mitochondria is a consequence of oxidative phosphorylation related to the respiratory chain [67]. ROS acts physiologically as signaling molecules, but when produced in abundance has deleterious consequences to DNA, protein, lipids, initiation of inflammatory events, excitotoxicity, and apoptosis [6]. 

Mitochondrial dysfunction, the increase in ROS and reactive nitrogen species (RNS), is involved in the pathogenesis of chronic neuropathic pain [68], leading to the degeneration of primary afferents [69], and ROS accumulation originates a feed-forward mechanism of nitroxidative lesion that triggers proapoptotic factors [70,71]. 

In fact, mitochondrial dysfunction is observed in various neuropathic pain phenotypes, such as chemotherapy-induced neuropathy, diabetic neuropathy, HIV-associated neuropathy, Charcot–Marie–Tooth neuropathy, and trauma-induced painful mononeuropathy [71]. 

More recently, it was shown that damage to mitochondria and aberrant mitochondrial transport in peripheral neurons are common features of peripheral neuropathy [72]. The complex and highly energy-consuming processes of neurotransmission by peripheral neurons are critically dependent on mitochondrial metabolic functions [73]. Moreover, peripheral neurons are primarily reliant on axonal transport of mitochondria, which facilitates the rapid movement of mitochondria to areas of high-energy demand along axons up to one meter in length. A key regulator of both mitochondrial movement and function is the epigenetic modifier histone deacetylase 6 (HDAC6) [72]. The accumulated evidence shows that HDAC6 inhibition is strongly associated with alleviating peripheral neuropathy and neuropathic pain, and mitochondrial dysfunction in vivo and in vitro models of peripheral neuropathy [72]. Although it is accepted that free radicals and the altered lipids and proteins have a crucial role in developing peripheral and central sensitization, the fine molecular mechanisms regulating their action in pain processing have not been elucidated [24] in man.

Some experimental studies demonstrated that the increase of ROS levels induces nociception [74,75]. However, removing excessive ROS or RNS by free radicals scavengers, a decrease in neuropathic and inflammatory pain is achieved [7,76,77]. 

Concerning the synapses at the spinal cord level, laboratory studies demonstrated that ROS increases excitatory signaling [78] using several pathways, including NMDA receptor activation [77] and the alteration of the proteins involved in glutamatergic homeostasis [79]. ROS-induced pain sensitivity that activates NMDA receptors increases Ca^2+^ influx to the cytoplasm, increasing mitochondrial ROS production and pain enhancement of neurons [7]. In contrast, there is a decrease in inhibitory transmission due to a reduction of Gamma-Aminobutyric Acid (GABA) release or death of GABAergic neurons [26].

ROS can also activate directly and indirectly the Ca^2+^-permeable channels [23], members of the TRP family, known to be involved in neuropathic pain [24,29]. Members of the subfamilies A (TRAP1), M (TRPM2 and 7), and V (TRPV1 and 4) have been demonstrated to play a role in nociception mediated by sensory neurons [24]. These channels are expressed in primary afferent nociceptors, abundant in the skin, and related neurons [24,80]. TRP channels allow calcium to enter the nerve terminals, activating apoptotic mechanisms leading to nerve terminal degeneration [81]. 

Finally, oxidative stress is involved in neuroinflammation by modulating neuroinflammatory signaling involving the NFKB, MAPK [82,83], TNF-α [83], and nuclear factor erythroid 2 (NFE2)-related factor 2 (NRF2) signaling pathways [84]. NRF2 is a transcription factor encoded by the *NFE2L2* gene that remains bound to Kelch-like ECH-associated protein 1 (KEAP1) in the cytoplasm that ultimately leads to proteasomal degradation. During peripheral neuropathy, NRF2 can translocate to the nucleus and binds to antioxidant response elements (AREs), leading to the transcription of several antioxidative enzymes and cytoprotective proteins that can ameliorate neuropathy and neuropathic pain in rodent models. Thus, modulating NRF2 activity is a promising pharmacological approach in inflammatory and painful diseases [85,86,87].

Rodrigo Sandoval et al. (2018) demonstrated that TNF-α increased ROS production in nociceptive neurons and p35 expression followed by TRPV1 phosphorylation and an increase in Ca^2+^ influx in nociceptive neurons and increased pain sensation, which may be targeted therapeutically [83].

### 3.5. Inflammation

For five decades, apoptosis has been recognized as fundamental for almost all the critical biological processes that occur in multicellular organisms, since embryogenesis and throughout life, to maintain the homeostatic balance of tissues and organs as well as the progression and final resolution of inflammation [88,89,90].

However, this physiological form of cell death was very early on associated with the absence of an inflammatory reaction by the surrounding tissues. This was initially explained by the preservation of membrane integrity in such a way that cellular content does not come into contact with neighbouring cells [87,88,91,92,93,94].

Inside, they contain well-preserved organelles and nuclear fragments, which will be prompt digested and removed from tissues by professional phagocytes or non-professional neighboring cells that specifically recognize apoptotic cells [87,92].

Although initially it was considered that the maintenance of the integrity of the cellular membrane was the only reason why apoptosis did not cause inflammation, it is now known and consensual that the mechanisms involved in the recognition and subsequent phagocytosis and clearance of apoptotic cells will determine whether apoptosis is immunologically “silent” or anti-inflammatory [93,94,95,96,97,98].

It seems clear that during apoptosis, the inflammatory response could be blocked [98] Regarding biochemical explanation, the fact that the inflammatory response is absent during apoptosis was initially explained by the exposure of phagocytes to phosphatidylserine after a process of translocation from the inner to the outer leaflet of the cell membrane ([99] This “eat me” signal that attracts and allows the recognition of apoptotic cells and their subsequent elimination plays a central role in phagocytosis and is one of the hallmarks of apoptosis [97,100,101].

Caspase activation, changes in mitochondrial membrane potential, and DNA cleavage are also players in apoptosis, as they contribute to disrupt cellular functions and mark cells for a “silent” phagocytic clearance [102,103].

Due to activation of caspases that cleave membrane channel proteins results in the release of adenosine monophosphate (AMP) [103].

Then, through the 5′-nucleotidase a phosphate group is removed from the AMP to produce adenosine that binds to the A2a receptors on the surface of phagocytes to generate the anti-inflammatory response [103].

Currently, other molecules, such as lysophosphatidylcholine, ATP and uridine 5’-triphosphate (UTP), sphingosine-1-phosphate, CX3CL1/fractalkine, thrombospondin-1, monocyte chemoattractant protein-1 (MCP-1), lactoferrin, are also known as effective in the “silent” clearance of apoptotic cells [104] These molecules act by sending chemotactic signals that attract phagocytes that recognize apoptotic cell-associated molecular patterns (ACAMPs) on the surface of apoptotic cells, therefore contributing to the anti-inflammatory characteristic of apoptosis. Besides, apoptotic cells can stimulate macrophages to generate anti-inflammatory mediators such as IL-10 or TGF-β, prostaglandin E2, and platelet activating factor (PAF) [105,106,107,108,109].

Considered as a type of physiological cell death, apoptosis also has a vital role in the responses to tissue injury or infection, determining the tissue’s ability to repair and regenerate after injury [97,110,111,112].

It is also important in controlling the number of inflammatory cells present at sites of inflammation. The fact is that many factors and signaling pathways that are activated and intervene in the inflammatory response are also involved in the regulation of apoptosis, and vice versa [108,113,114]. 

Thus the concept that apoptosis is non-inflammatory is not always a correct one [115] For example, it was observed in an experimental model that the injection of agonistic antibody against the Fas receptor causes hepatocytes apoptosis and inflammation response. In the same experimental model, the authors also observed that if they inhibited or prevented apoptosis by administering a caspase-3 inhibitor, hepatic inflammation would also be reduced [116]. These findings indicate that, at least, in this case, apoptosis and inflammation share signaling pathways and others signaling factors [108].

Although apoptosis does not cause an inflammatory response, inflammation does sometimes occur, which is a matter that is not yet fully clarified. Some studies suggest that this may be due to how quick the apoptotic process occurs and how quick and efficient apoptotic cells are removed by phagocytes since, if this clearance is not effective, over time, the apoptotic cells develop a process known as secondary necrosis [93,97,117].

In case of excess of apoptosis or failure in cleaning cells in secondary necrosis, the membranes are degraded and become permeable allowing the escape of intracellular contents with proinflammatory molecules, thus stimulating the inflammatory response [97].

This inflammatory response to apoptotic cells should be viewed with optimism and properly explored, as it can be helpful in certain pathologies in which apoptosis has an important role, such as some viral infections and cancer. In these cases, the existence of an acute inflammation, associated with or in response to the abundance of apoptosis, may play an important role in promoting tissue healing/repair [109,115,118,119].

The presence of unremoved apoptotic cells has been linked to several different diseases, such as some infections or diseases with a marked inflammatory background, in the course of which Pathogen-Associated Molecular Patterns (PAMPs) and Damage-Associated Molecular Patterns (DAMPs) are detected and elicit leukocytes aggregates to destroy the pathogen and to start the repair process [120]. After, leukocyte recruitment ceases, and those that had previously been recruited will be rapidly ingested by resident phagocytes and eliminated. In normal conditions, this clearance process can be remarkably efficient, and because of it, it is actually difficult to see apoptotic cells in organs and tissues. This failure can lead to a prolonged inflammatory response, causing apoptotic cells to accumulate locally, which often leads and has been observed in some diseases with an inflammatory background. This points to a tight link between cell death and inflammation [104,109,110,121].

In summary, apoptosis plays an important role in normal wound healing and tissue regeneration, and could fulfill the clinical need to prevent the harmful consequences of inflammation, for which apoptotic cells clearance is crucial. Finally, strategies to avoid defective clearance of apoptotic material could constitute new approaches for treating inflammatory or autoimmune diseases. 

### 3.6. The p38MAPK Pathway and Apoptosis

Numerous studies have shown that activation of Mitogen-activated protein kinases (MAPK), particularly c-Jun N-terminal kinase (JNK) and p38, contributes to neuropathic pain pathology [43,122,123,124].

The activation of the p38MAPK signaling pathway plays an essential role in the generation and maintenance of neuropathic pain by regulating transcription, protein synthesis, receptor expression, and inducing apoptosis. The protein kinase A (PKA) is a key protein involved in neuropathic pain signaling, which by activating the p38MAPK intervenes in the apoptosis of spinal cord cell [46]. Studies have shown that PKA is closely related to inflammatory pain and bone cancer pain [124] but it is not clear its involvement in neuropathic pain caused by nerve damage. Authors hypothesized that PKA is involved in neuropathic pain by mediating spinal cord cell apoptosis through p38MAPK pathway activation, which culminates in a decrease in the antiapoptotic protein BCL2 and an increase in the proapoptotic proteins, TNF-alfa, IL1-beta, BAX, Caspase3 and 9 (Figure 1) [46]. Further, it has been suggested that the inhibition of either JNK or p38 may represent a potent clinical target for neuropathic pain management [46,122].

## 4. Apoptosis and Clinical Implications

The deregulation of cell death contributes to the development of numerous pathologies, namely cancer, ischemia, autoimmune, neurodegenerative diseases, and pain. The changes in cell death in these pathologies are not the same. There is a characteristic increase in some of them, and in others, there is a decrease in apoptosis.

Neuropathic pain is a kind of pain caused by damage to somatosensory nervous system mediated by multiple risk factors, including toxicity, surgery, and trauma. It reduces the quality of life of patients, being considered a public health problem worldwide. In this context, it becomes evident the need to deepen and improve the knowledge of the pathogenesis and pathophysiology in order to identify new biomarkers for the treatment of neuropathic pain. 

### 4.1. Apoptosis and Neuropathic Pain

Nerve damage due to oxidative stress and mitochondrial dysfunction is a key pathogenic mechanism involved in chemotherapy-induced peripheral neuropathy (CIPN) [23,24,26,27,28,29,120]. On the other hand, TRP channels allow calcium to penetrate into nerve terminals. This can activate apoptosis mechanisms that result in nerve terminal destruction [33] in the skin which can cause long-term pain relief [33]. Further, cisplatin, oxaliplatin, and paclitaxel-induced mitochondrial oxidative stress, inflammation, cold allodynia, and hyperalgesia, through an increase in TRPA1 and TRPV4, leading to Ca^2+^ influx through direct channel activation or excessive production of oxidative stress and induction of apoptosis. The pain resulting from this Ca^2+^ overload is mediated through substance P (SP) and excitatory amino acid production. This chemotherapy-induced oxidative stress in DRG neurons that contribute to peripheral pain may be prevented with TRPA1 and TRPM8 antagonists such as reduced glutathione (GSH) and selenium [11,24,28,55,125,126].

Several studies have demonstrated that apoptotic activities in injured dorsal root ganglia increased after injury in different rat models [42,127,128] and that some agents could simultaneously suppress pain behavior and apoptotic activities [129,130]. However, there is no direct evidence that agents resulting in inhibition of apoptotic activities in the injured neurons could attenuate pain behavior [35].

### 4.2. Biomarkers and Circulating Mediators in Pain

Oxidative damage to peripheral neurons can cause damage to the myelin sheath, mitochondrial proteins, and other antioxidant enzymes [33]. Identification of levels of malondialdehyde, glutathione (GSH), superoxide dismutase (SOD), and activities of mitochondrial enzymes such as citrate synthase and ATP synthase can help monitor the course of peripheral neuropathy and response of neuropathy to the treatment [33].

Previous studies have indicated that proinflammatory cytokines, such as TNF-α, play an important role in the peripheral mediation of neuropathic pain and that altered dorsal root ganglion (DRG) function and the degree of DRG neuronal apoptosis are associated with spinal nerve injury. The level of TNF-alpha expressed in DRG after distal nerve crush injury is higher than that after proximal crush injury. The expression of DRG apoptosis in the distal crush injury was higher than in the proximal crush injury [42]. Nerve root crush injuries proximal to the DRG, such as those that may occur in the cauda equina, are less painful than spinal nerve crush injuries. The mechanism for these results may relate to DRG TNF-alpha expression and apoptosis [34,42,74,129,130]. 

Various stimuli, such as oxidative stress, TNF-α, and FAS antigen activation, can the phosphorylate serine/threonine kinase domain and activate Apoptosis signal-regulating kinase 1 (ASK1), an upstream protein in the p38 and JNK pathways. ASK1 has an important role in neuroinflammation during the induction and maintenance of chronic pain. Therefore, besides its potential as a pain biomarker, the inhibition of ASK1 may be a new therapeutic approach for neuropathic pain [131].

A study developed by Yuanzhi Peng and colleagues [62] shows that the upregulation of Ube2b, ubiquitin-conjugating enzyme E2B (Ube2b) as previously mentioned, ameliorates neuropathic pain by regulating potassium voltage-gated channel subfamily A member 2 (Kcna2) in primary afferent neurons, suggesting that this might be a novel biomarker for neuropathic pain-targeted therapies. This new knowledge opens perspectives for new therapeutic targets in various diseases, namely chronic pain [62]. 

Several recent studies have found that ubiquitin system failure is implicated in chronic pain, mainly including neuropathic pain (NP) and inflammatory pain [57]. Inflammatory pain is caused by an increase in the excitability of peripheral nociceptive fibers due to changes in ion channel activity caused by inflammatory mediators. Pain is triggered by normally innocuous stimuli during the inflammatory process and becomes chronic if the inflammation is not resolved. The mechanisms of NP are partly distinct from those of inflammatory pain, and potential therapeutic targets mediated by the ubiquitin system are different in NP and inflammatory pain [57].

### 4.3. Pain Therapy through Modulating Apoptosis Activities

Only one-third of patients receive pain relief from current analgesics, such as opiates, nonsteroidal anti-inflammatory drugs, local anesthetics, tricyclic antidepressants, and anticonvulsants, including carbamazepine and gabapentin. Therefore, it seems notable that recent developments in understanding the mechanisms that produce pain, either individually or collectively, have disclosed new potential therapeutic targets for developing more effective drugs [1].

The identification of antioxidant molecules with pleiotropic activity in other pathophysiological pathways involved in the CIPN could aid in the development and improvement of new therapies [120].

Isoflurane, a general inhalation anesthetic used for the induction and maintenance of general anesthesia, promotes PI3K/AKT activation, upregulates B-cell lymphoma 2 (Bcl-2)-associated X protein Bcl-2 expression levels, and reduces the expression levels of caspase 3 and caspase 8 in myocardial cells. Isoflurane is beneficial for pain attenuation and inhibits apoptosis in myocardial cells via the PI3K/AKT signaling pathway in mice during cardiac surgery [132]. Further, Masahiko Kawaguchi et al. (2004) showed that isoflurane reduced the early development of neuronal apoptosis in rats subjected to focal cerebral ischemia [133]. Other studies demonstrated that low concentrations of isoflurane attenuated the Aβ-induced reduction in Bcl-2/Bax ratio and led to a mild elevation in cytosolic calcium levels. However, these results are dose-dependent, suggesting that isoflurane may have dual effects on Aβ-induced toxicity (protection or promotion) [134]. JNK and p-JNK were found to continually increase with prolonged AOPPs stimulation. They were activated most significantly under the treatment of 50 µg/mL AOPPs–RSA. By using a JNK inhibitor (SP600125), DRG neurons were protected from AOPPs stimulation [34]. 

Paeoniflorin can not only significantly inhibit the activation of ASK1 and simulate the analgesic effect of ASK1 inhibitors, but also significantly inhibit the response of glial cells and neuroinflammation induced by CCI [131]. Acting in the same pathway, administration of a p38 inhibitor and a JNK inhibitor ameliorates neuropathic pain symptoms in rodent models [135,136].

Spare nerve injury (SNI) induces a significant increase in PKA expression in the spinal cord, and PKA is involved in neuropathic pain by activating the p38MAPK pathway to mediate spinal cord cell apoptosis [46].

Granulocyte colony-stimulating factor (G-CSF) is a cytokine that promotes the survival, proliferation, and differentiation of neutrophils. Recent studies have indicated that G-CSF also has non-hematopoietic functions and can potentially be used for the treatment of neuronal injury, including stroke and neurodegenerative disease. A clinical trial has shown that G-CSF may have a therapeutic effect on spinal neuropathic pain [137]. Several mechanisms and different signaling pathways are involved in the therapeutic role of G-CSF in neuropathic pain, including the upregulation of Mu-opioid receptor (MOR) and autophagic activity, through the suppression of BCL-2 expression in the injured nerve in the early phase [35]. 

Tetramethylpyrazine (TMP) attenuated neuropathic pain-associated hyperalgesia and neuronal apoptosis in the spinal dorsal horn, which was demonstrated by a decreased number of TUNEL-positive cells, upregulation in BCL-2 expression, and downregulation in caspase-3 expression in the spinal dorsal horn. These results suggest that TMP is beneficial for treating neuropathic pain by inhibiting apoptosis via the modulation of BCL-2 and caspase-3 proteins [138].

The crucial role of NF-kB in several pathologies that accompany pain provides evidence that targeting the NF-kB signaling cascade, including UPP, might have beneficial antinociceptive effects [64,65,66]. Several proteasome inhibitors are already approved to treat hematological neoplasias, such as multiple myeloma [139]. The generation of new proteasome inhibitors may represent a new pharmacotherapy for inflammatory pain [57]. Several studies show that UPP inhibitors can prevent inflammatory pain following injury and infection. The proteasome inhibitor MG132 can inhibit the activation of NF-κB, reversing the inflammatory pain [140].

Drugs that neutralize TNF have been developed and are used clinically to treat inflammatory and autoimmune diseases, such as rheumatoid arthritis, inflammatory bowel disease, and psoriasis [141]. However, despite their clinical success, the use of anti-TNF drugs is limited, but antibodies and bio-engineered ligands are currently in the preclinical and early clinical stages of development (such as infliximab, adalimumab, golimumab, and several biosimilars). Preclinical data obtained in different disease models show that the selective targeting of TNFRs has therapeutic potential and may be superior to global TNF blockade in several disease indications [141].

Moreover, studies have demonstrated that chemotherapy drugs, such as vincristine (Vin), could cause neuropathic pain and mitochondrial dysfunction. The mitochondrially targeted antioxidant, mitoquinone (MitoQ), can modify mitochondrial signaling, resulting in beneficial effects on various diseases. Chen et al. investigated whether MitoQ could regulate vincristine-induced neuropathic pain and the underlying molecular mechanisms. They observed reduced expression of cleaved caspase 3 and BAX and increased protein levels of the antiapoptotic factor BCL-2 in the spinal cord of MitoQ-treated mice after vincristine stimulation. Further, MitoQ could alleviate vincristine-induced neuropathic pain by inhibiting oxidative stress and apoptosis via the amelioration of mitochondrial dysfunction (Figure 3) [142].

Joseph et al. demonstrated that the caspase inhibitors could decrease the pain behaviors of rats that received chemotherapy (vincristine) and HIV therapy drugs [143]. Scholz et al. also showed that the caspase inhibitor could decrease pain behavior and the apoptosis of the dorsal horn of rats that received partial peripheral nerve injury [144].

There is also scientific evidence showing that antioxidants may be used preventively and therapeutically not only to reduce oxidative stress-related parameters, but also inflammatory response and pain in several diseases [11,145,146], and these effects could contribute to the reduction in apoptosis observed in neuropathic pain.

A randomized controlled study reported that L-carnosine exerted neuroprotective activity by significantly decreasing proinflammatory (NF-κB, TNF-α) and apoptotic (caspase 3) markers and increasing Nrf2 in colorectal cancer patients treated with oxaliplatin-induced PN. Therefore, the use of Nrf2 inducers in treating neuropathic pain in clinical settings holds great promise in the future [84].

The use of conventional analgesics, such as nonsteroidal anti-inflammatory drugs, opioids, tricyclic antidepressants, and anticonvulsants are reported to exhibit a wide spectrum of adverse effects, some of which limit their use for the treatment of neuropathic pain. Thus, the identification of new therapeutics with minimal side effects that can be used to alleviate neuropathic pain is important. For this purpose, the research and identification of analgesic molecules from natural sources could be considered. For instance, flavonoids and other antioxidants may attenuate neuropathic pain in different models summarized in Figure 3 [11,147]. 

Moreover, there are several ongoing preclinical studies targeting apoptosis mechanisms in order to treat chronic pain (Table 1). However, future studies are needed to translate these results into clinical practice. 

## 5. Conclusions and New Perspectives

The deregulation of apoptosis is involved in the development of several pathologies, and recent evidence suggests that apoptosis may be involved in chronic pain, namely neuropathic pain. 

Neuropathic pain is a chronic pain condition caused by primary damage or dysfunction of the nervous system, but the fine molecular mechanisms have not yet been fully elucidated.

Previous studies have indicated that NF-kB, proinflammatory cytokines such as TNF-α, and oxidative stress play an important role in peripheral mediation of neuropathic pain through apoptosis modulation. Further, the degree of DRG neuronal apoptosis has recently been associated with spinal nerve injury via caspase signaling and/or PKA activation through the p38MAPK pathway, generating and maintaining neuropathic pain. 

The adoption of measures to prevent and protect mitochondrial function may constitute a promising therapeutic strategy to alleviate or prevent chronic pain conditions [79] since mitochondria are a primary source of cellular ROS [120]. Pharmacological interventions aimed at maintaining normal mitochondrial function may be an alternative therapeutic approach to the direct free radical scavengers for the treatment of CIPN [120]

Medications that can modulate proinflammatory cytokine expression, the NF-kB pathway, and apoptosis, such as TNF-alpha modulators, G-CSF, proteasome inhibitors, and flavonoids, among others, have the potential to treat neuropathic pain [35]. However, the timing of administration, dose, and indications for neuropathic pain treatment still need to be validated by clinical studies [35].

Further, the inhibition of apoptosis via the modulation of BCL-2 and caspase-3 is beneficial for the treatment of neuropathic pain [138].

The involvement of apoptosis in pain and the clinical relevance of this knowledge in the potential discovery of new biomarkers and therapeutic targets can result in the development of more effective and targeted drugs to treat chronic pain, particularly neuropathic pain.

## Figures and Tables

**Figure 1 biomedicines-10-01255-f001:**
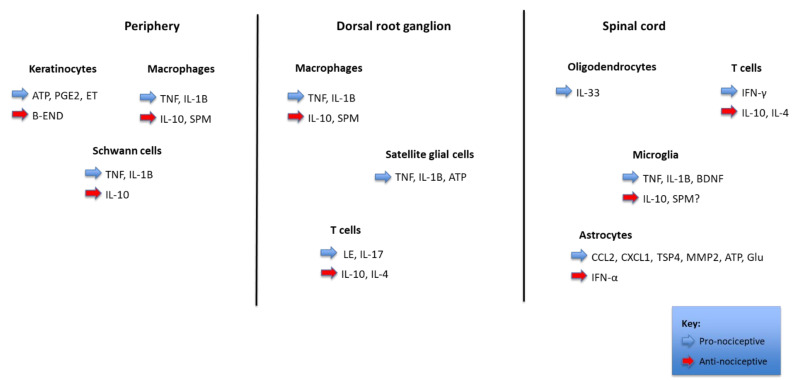
Pain regulation by non-neuronal cells and inflammation. The figure shows the interactions between distinct parts of a nociceptor (periphery, dorsal root ganglion, and spinal cord) with different types of non-neurons cells (keratinocytes, Schwann cells, satellite glial cells, oligodendrocytes, and astrocytes), immune cells (macrophages, T cells, and microglia), cancer cells, and bone marrow stem cells. These cells produce pronociceptive (highlighted in blue) and antinociceptive (highlighted in red) mediators, which modulate the nociceptor sensitivity and excitability through binding to their respective receptors. In the spinal cord dorsal horn, the central terminal of the nociceptor forms a nociceptive synapse with a postsynaptic neuron to mediate pain transmission in the Central Nervous System (CNS).

**Figure 2 biomedicines-10-01255-f002:**
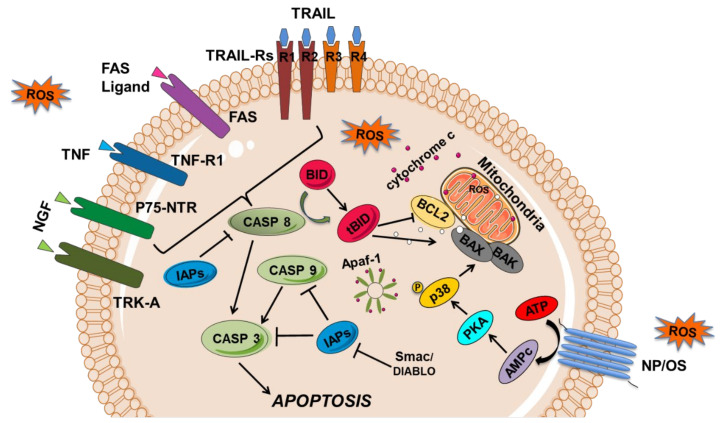
Representation of cell signaling pathways leading to apoptosis. Apoptosis can result from the activation of the extrinsic and intrinsic apoptosis pathways. The extrinsic or membrane pathway starts by the interaction of a death ligand from the TNF (tumor necrosis factor) family, TNF, FAS-L, TRAIL (TNF related apoptotic inducing ligand), and NGF (nerve growth factor) to the respective receptors, TNF-R1, FAS, TRAIL-R1-R2, and p75-NTR, which subsequently activate a caspase cascade (caspase 8 and 3). In mitochondria-mediated apoptosis or intrinsic pathway, proteins of BCL-2 family (BAX, BAK; BCL2) and cytochrome c are released, binding to the apoptotic protease-activating factor-1 (APAF-1) and caspase 9 (CASP 9) forms a complex called the apoptosome, which is capable of recruiting caspase-9, -3 (CASP3) and -7, and inducing apoptosis. Several apoptotic regulators are also represented, such as the CASP inhibitors (namely the IAPs—inhibitor of apoptosis proteins) and the IAPS inhibitors SMAC/DIABLO (second mitochondria-derived activator of caspase/direct inhibitor of apoptosis-binding protein with low pI). In neuropathic pain (NP), oxidative stress (OS)/ROS generated exogenously or endogenously can activate p53, which activates the proapoptotic BCL-2 proteins that can inhibit the functions of the antiapoptotic proteins, such as BAX and BAK. Besides, OS can induce nerve apoptosis by activating BAX through protein kinase A (PKA) and p38 pathway. Transmembrane death receptors such as FAS, TRAIL-R1/2, and TNF-R1 can also be activated by ROS.

**Figure 3 biomedicines-10-01255-f003:**
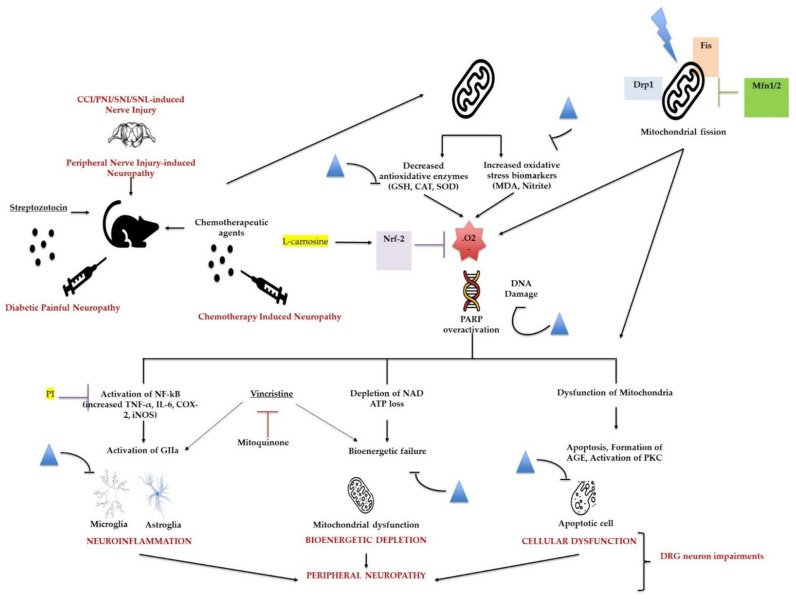
The effects of flavonoids and mitoquinone on neuropathic pain induced by vincristine. Flavonoids attenuate different peripheral neuropathic pain conditions by inhibiting or downregulating different neuroinflammatory, cellular, bioenergetic, and oxidative stress markers. Mitoquinone alleviates vincristine-induced neuropathic pain by improving mitochondrial dysfunction and inhibiting oxidative stress and apoptosis. Flavonoids; Nrf-2: nuclear factor erythroid 2–related factor 2; Drp-1: dynamin-related protein 1, Fis: mitochondrial fission protein; Mfn: mitofusin; .O2: reactive oxygen species/superoxide anion; PARP: poly(ADP-ribose) polymerase; NAD: nicotinamide adenine dinucleotide; AGE: advanced glycation end products; PKC: protein kinase C; PI: proteasome inhibitor.

**Table 1 biomedicines-10-01255-t001:** Some preclinical/clinical studies with drugs/compounds involved in apoptosis that could potentially treat pain.

Drugs	Administration and Time of the Experiment	Main Results	Model	Localization of Changes	References
**Apocynin**	Intrathecal lumbar injection; studied 42 days after injury	↓ NP, mechanical sensitivity	Animal model of SCI—Rat	Behavioral testing only	[145]
**Corilagin**	IP injection for 4 weeks after injury	↓ Oxidative stress, NF-ƘB, ↓ TNF-α, IL6 ↓ Caspase 3, ↓ Pain	Animal model of SCI—Rat	Spinal cord	[148]
**Curcumin**	IP injection for 28 days after injury	↓ IL-2, NF-ƘB, TNF-α ↓ Pain	Animal model of SCI—Rat	Spinal cord	[149]
**Deferoxamine**	IP injection on the first day after injury, followed by once a week during 12 weeks	↓ Oxidative stress ↓ Iron overload markers ↓ Pain	Animal model of SCI—Rat	Hind limb sensory cortex, hippocampus, thalamus	[150]
**Epigallocatechin 3-gallate**	IV infusion for 36 h after 4 h post-lesion (acute) and after 12 months (chronic)	↓ Glial activity ↓ NP ↓ Lesion size area ↑ Number of spinal neurons	Animal model of SCI—Rat	Spinal cord	[151]
**Geraniol**	IV infusion 6h after lesion; evaluation at 4 weeks and 8 weeks	↓TNF-α, ↓NMDAR1 ↓ Caspase 3, glial activation ↑ HO-1, Nrf2. ↓ NP	Animal model of SCI—Rat	Spinal cord	[152]
**Acupuncture**	Once a day on 31 to 33 d after lesion	↓ Inflammation inhibitor, ROS scavenger, ↓ ERK inhibitor, ↓ p38MAPK inhibitor ↓ NP	Animal model of SCI—Rat	Spinal cord (L4–L5)	[153]
**Progesterone**	Daily sc injections (6 h, 1 day, 28 days after injury)	↓ NF-ƘB, COX-2, iNOS ↓ NP	Animal model of SCI—Rat	Spinal cord (L4–L5)	[154]
**SB203580** **(P38MAPK antagonist)**	35 days Intraspinal injection in L4/L5 space	SCI ↑-P38MAPK ↓ P38MAPK, NP with antagonist	Animal model of SCI—Rat	Spinal cord (dorsal horn neurons)	[155]
**Resiniferatoxin** **(molecular neurosurgery)**	One time	↓ TRPV1 ↓ NP, cancer pain	Clinical trial phase 2		[156]
**Hypericum perforatum**	Gastric gavage for 4 weeks	↓ TRPV1, TRPM8, ROS ↓ NP	Animal model of CCI injury of the sciatic nerve—Rat	Sciatic nerve and DRG	[157]
**VAS2870 (NADPH oxidase inhibitor)** **Roscovitine (CdK5 activity inhibitor)**	24 h 1 h	↓NOX1, NOX2/NADPH oxidase complex, TNF-α ↓Inflammatory pain	Primary culture of mouse nociceptive neurons	Trigeminal ganglia, DRG, and HEK293 cells	[83]
**Isoflurane (low concentration)**	One time (6 h or 30 min)	↑ PI3K/AKT activation ↑Cytosolic calcium levels	Cell line Mice primary neurons	Mice primary neurons H4 human neuroglioma cells	[134]
**SP600125** **(JNK inhibitor)**	One time	AOPP ↑ JNK, and ↑ DRG neurons apoptosis by activating caspase 3 and PARP-1. ↓ by SP600125	Cell culture Rat model	Rat DRG neurons	[34]
**Paeoniflorin** **(ASK1 inhibitor)**	One time	↓ ASK1, glial cells, and neuroinflammation, ↓ ↓ ↓ TNF-α, IL-1	Animal model of CCI injury of the sciatic nerve—Rat	Spinal dorsal horn	[131]
**G-CSF**	One time	↑ MOR, autophagic activity, BCL-2	Clinical trial		[137]]
**Tetramethylpyrazine**	One time (3, 7 and 14 days post-surgery)	↑ BCL-2 ↓ Caspase-3 expression ↓ NP	Animal model of CCI injury of the sciatic nerve—Rat	Superficial spinal dorsal horn (L4–L5)—laminae I and II	[138]
**MG132** **(Proteasome inhibitor)**	One time	↓ Activation of NF-κB ↓ Inflammatory pain	Rat model of rheumatoid arthritis and SCI	DRG	[140]
**Mitoquinone**	One time	↓ Caspase-3 and BAX ↑ BCL-2 ↓Vincristine-induced neuropathic pain	Vincristine-induced neuropathic pain model in ICR mice Primary neuron culture of DRG	Lumbar (L4–L5) DRG Spinal cord (L4–L5) Sciatic nerve	[142]

↓ Decrease, ↑ Increase. DRG: dorsal root ganglia; COX-2: cyclooxygenase-2, HO-1: heme oxygenase-1, IL: interleukin, iNOS: inducible NO synthase, ERK: extracellular-signal-regulated kinase, MMP: matrix metalloproteinase, mTOR: mammalian target of rapamycin, NF-κB: nuclear factor-κB, Nrf2: nuclear factor erythroid 2-related factor 2, NP: neuropathic pain, PI3K: phosphoinositide 3-kinases, p38MAPK: p38 mitogen-activated protein kinases, ROS: reactive oxygen species, TNF-α: tumor necrosis factor-alpha. SCI: spinal cord injury, CCI: chronic constriction injury, IP: intraperitoneal, IV: intravenous, SC: subcutaneous, NMDAR1: N-methyl-D-aspartate receptor 1, NP: neuropathic pain, SCI: spinal cord injury, DRG: dorsal root ganglion, AOPP: advanced oxidative protein products, ASK1: apoptosis signal-regulating kinase 1, G-GCF: granulocyte-colony-stimulating factor (G-CSF).

## Data Availability

Not applicable.

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
