# Peer review of "Apoptosis and (in) Pain—Potential Clinical Implications"

_biomedicines, 2022, doi:10.3390/biomedicines10061255_

Round 1
Reviewer 1 Report
This review by Hugo Ribeiro et al describes the pathways of apoptosis and its role in pain. The authors have nicely summarized the various pathways by which the “programmed cell death” can occur and its interplay with the mitochondria, proteasome system, unfolded protein response of the endoplasmic reticulum, the p38MAPK pathway, oxidative stress, and inflammation. The authors have described the clinical implications of apoptosis with specific emphasis on “pain”. I have some minor comments:
- In figure 1 “Bone marrou stem cells” can be changed to “Bone marrow stem cells”
- On page 4 “serie” should be changed to “series”
- On page 8 “inhiting” should be changed to “inhibiting”
- In figure 3 “Miroquinone” should be changed to “Mitoquinone”
- The sentence on page 9
“However, removing excessive ROS or RNS by free radicals scavengers decrease in neuropathic and inflammatory pain is achieved”
Can be modified to
“However, by removing excessive ROS or RNS using free radicals scavengers, a decrease in neuropathic and inflammatory pain is achieved”
- The sentence on page 7
“The change in mitochondria membrane potential with the opening of the MTPP leads to the release into the cytoplasm of cytochrome c, a mitochondrial inner membrane protein, which induces cell death by apoptosis.”
Can be modified to
“The change in mitochondria membrane potential with the opening of the MTPP leads to the release of cytochrome c, a mitochondrial inner membrane protein, into the cytoplasm which induces cell death by apoptosis.”
Author Response
Reviewer 1 Comments
We thank you for taking the time to assess our manuscript and for the constructive comments and text corrections. We have addressed all your comments and corrections. In the revised manuscript, we highlight the improved text using the “Track Changes” function.
Reviewer 2 Report
The concept of Apoptosis being involved in pain, including neuropathic and chronic, and that targeting apoptosis pathways can alleviate pain is interesting and the review would be of interest to the readers of Biomedicine. Please address the following comments prior to publication.
- The references appear to not always be original references, but instead referencing later papers or reviews discussing the research of interest. Please update references to original references throughout the manuscript.
- The font size in Figure 1 and 4 is too small to read and needs to be increased by a lot.
- Page 6, insert reference for NGF. Following first sentence on the page.
- Page 9, towards the bottom, it should be inhibiting, not inhiting.
- Page 8, bottom of page, what other pathways in addition to NFKB are of interest?
- Page 13, third paragraph, should be “Penga and colleagues”, not “ Penga and col”.
- Weird font size on page 14 and 16.
Author Response
Reviewer 2 Comments
We thank you for taking the time to assess our manuscript and for the constructive comments. We have addressed all your comments. Please, see below the answers to your specific comments. In the revised manuscript, we highlight the improved text using the “Track Changes” function.
Comment 1. The references appear to not always be original references, but instead referencing later papers or reviews discussing the research of interest. Please update references to original references throughout the manuscript.
Answer:
The articles that we cited in our work are the ones that we actually consulted and studied to write our review article. So, in fact, they are not always the original references. We believe that changing what we did and referencing different articles/works will not be the most correct. We ask you to consider our position as it seems to us to be the most honest. On the other hand, future readers who need to access the original studies can do so easily through the references that we have placed in the article.
Comment 2.
The font size in Figure 1 and 4 is too small to read and needs to be increased by a lot.
Page 6, insert reference for NGF. Following first sentence on the page.
Page 9, towards the bottom, it should be inhibiting, not inhiting.
Page 8, bottom of page, what other pathways in addition to NFKB are of interest?
Page 13, third paragraph, should be “Penga and colleagues”, not “ Penga and col”.
Weird font size on page 14 and 16.
Answer:
We have increase the font size of the figures the maximum as possible.
The NGF reference has been inserted (Lewin & Nykjaer, 2014).
We have corrected inhibiting.
We have added other interest pathways in addition to NFKB, as TNF-a (Rodrigo Sandoval et al., 2021) and Nuclear factor erythroid 2 (NFE2)-related factor 2 (NRF2) signaling pathways (Ahmed et al., 2017, Larissa Staurengo-Ferrari et ., 2019, Paramita Basu et al., 2022).
We have corrected “Penga and colleagues” and the font size
Reviewer 3 Report
In this article, Ribeiro et al. reviewed the role of apoptosis in pain formation. Although the evidence of the involvement of apoptosis in pain is less than autophagy, which is another important mechanism when cells encounter stress, this topic remains an important field of pain research and is worth having a detailed review. This article had three main parts. In the first part, the authors tried to review the pathophysiology of neuropathic pain development. In the second part, the authors reviewed the mechanisms of the apoptotic process including the ubiquitin system and oxidative stress. In the third part, the author reviewed the role of apoptosis in chronic neuropathic pain formation.
Generally speaking, the authors tried to have a detailed review of this topic. However, the first and second parts of this article are too long and a little redundant, and the authors need to have more discussions on the third part of this article, which is the most important part of this review. The authors had a novel introduction of apoptosis involved in chronic neuropathic pain formation; they introduced three mechanisms including pro-inflammatory cytokines, TRP channels, and oxidative stress. However, it is pity that the author did not have detailed discussions of the studies about the roles of TRP channels and oxidative stress in the apoptosis in chronic pain formation. The authors also seem to over-explain some results of different studies. There are some suggestions before publication.
- As mentioned before, there is only a little direct evidence that showed suppression of apoptosis activities could decrease pain behavior. As I know, Joseph et al. demonstrated that the caspase inhibitors could decrease the pain behaviors of rats that received chemotherapy (vincristine) and HIV therapy drugs [1]. Scholz et al. also showed that caspase inhibitor could decrease pain behavior and the apoptosis of the dorsal horn of the rats that received partial peripheral nerve injury [2]. The authors need to add these two references and have some discussion. However, many studies including Chen et al.’s research [3] which were cited by authors seem only could demonstrate that medication (mitoquinone) can suppress apoptosis activities and pain behavior at the same time, but did not provide direct evidence showing that inhibition of apoptosis could suppress pain behaviors. The authors may make more discussions about the direct and indirect evidence of apoptosis involvement in pain formation in detail.
- The parts “2. Pain – Definition and mechanisms – a brief review”, and “3. Apoptosis and pain” are a little too long and need to be condensed. The authors may only keep the contents that will be discussed and mentioned in “4. Apoptosis and clinical implications”.
- “4.1. Apoptosis and neuropathic pain” can be merged into “4.3. Apoptosis in Pain therapy”.
- In the description of “3. Apoptosis in Pain therapy…Isoflurane, a general inhalation anesthetic”, it seems a little weird that the authors explain the analgesic effects of isoflurane are through inhibiting apoptosis of myocardial cells. Is there evidence that isoflurane could inhibit apoptosis of neurons?
- Figures 3 and 4 described the mechanisms of two different individual drugs. The author may make a single figure which presents all apoptotic pathways and summarize all drugs involved in apoptosis and pain in this single figure.
- In “table 1. Some preclinical studies in pain with biomarkers linked with apoptosis”, the authors should cite the reference to the individual drugs. Some drugs in this table are not involved in the apoptotic pathway (ex: rapamycin is a medication that can promote autophagy). The authors may create a new table that listed all drugs which were discussed in apoptosis and pain formation in this review.
- The references style is not the same as the style suggested by biomedicines.
Reference:
- Joseph, E.K.; Levine, J.D. Caspase signalling in neuropathic and inflammatory pain in the rat. Eur J Neurosci 2004, 20, 2896-2902, doi:10.1111/j.1460-9568.2004.03750.x.
- Scholz, J.; Broom, D.C.; Youn, D.H.; Mills, C.D.; Kohno, T.; Suter, M.R.; Moore, K.A.; Decosterd, I.; Coggeshall, R.E.; Woolf, C.J. Blocking caspase activity prevents transsynaptic neuronal apoptosis and the loss of inhibition in lamina II of the dorsal horn after peripheral nerve injury. J Neurosci 2005, 25, 7317-7323, doi:10.1523/JNEUROSCI.1526-05.2005.
- Chen, X.J.; Wang, L.; Song, X.Y. Mitoquinone alleviates vincristine-induced neuropathic pain through inhibiting oxidative stress and apoptosis via the improvement of mitochondrial dysfunction. Biomed Pharmacother 2020, 125, 110003, doi:10.1016/j.biopha.2020.110003.
Author Response
Reviewer 3 Comments
We thank you for taking the time to assess our manuscript and for the constructive comments. We have addressed all your comments. Please, see below the answers to your specific comments. In the revised manuscript, we highlight in yellow the improved text using the “Track Changes” function.
Comment 1. However, it is pity that the author did not have detailed discussions of the studies about the roles of TRP channels and oxidative stress in the apoptosis in chronic pain formation.
Answer: In part 4.1 we add some text to fill in your suggestion.
Recently, 9 TRP channels (TRPA1, TRPC5, TRPM2, TRPM4, TRPM7, TRPV1, TRPV2, TRPV3, and TRPV4) had been demonstrated to be activated by oxidative stress (Mori et al., 2016) and in neurons related to nociception the expression levels of four TRP channels (TRPA1, TRPM2, TRPV1 and TRPV4) are high (Cristina Carrasco1 et al., 2021).
TRPV1 (Bourinet et al., 2014) and TRPA1 has been demonstrated in neuropathic pain associated to diabetes or the administration of chemotherapeutics, probably mediated by the synthesis of reactive oxygen and nitrogen species (Kim and Hwang, 2013; Huang et al., 2017), which are well-known TRPA1 activators (Trevisan et al., 2016, Raquel Diez et al., 2021).
… Further, cisplatin, oxaliplatin, and paclitaxel induced mitochondrial oxidative stress, inflammation, cold allodynia, and hyperalgesia, through an increase in TRPA1 and TRPV4, leading to Ca2+ influx through direct channel activation or excessive production of oxidative stress and induction of apoptosis. The pain resulting from this Ca2+ overload is mediated through substance P (SP) and excitatory amino acid production. This chemotherapy-induced oxidative stress in DRG neuron that contribute to peripheral pain may be prevented with TRPA1 and TRPM8 antagonists as Reduced Glutathione (GSH) and selenium (Materazzi et al., 2012, Kahya et al., 2017, Mustafa Nazıroglu et al., 2021).
Comment 2. As mentioned before, there is only a little direct evidence that showed suppression of apoptosis activities could decrease pain behavior. As I know, Joseph et al. demonstrated that the caspase inhibitors could decrease the pain behaviors of rats that received chemotherapy (vincristine) and HIV therapy drugs [1]. Scholz et al. also showed that caspase inhibitor could decrease pain behavior and the apoptosis of the dorsal horn of the rats that received partial peripheral nerve injury [2]. The authors need to add these two references and have some discussion. However, many studies including Chen et al.’s research [3] which were cited by authors seem only could demonstrate that medication (mitoquinone) can suppress apoptosis activities and pain behavior at the same time, but did not provide direct evidence showing that inhibition of apoptosis could suppress pain behaviors.
Answer: We added the two suggested references (Joseph et al. and Scholz et al) and the purposed text. We also added some text containing studies that could show that inhibition of apoptosis could suppress pain behaviors (Campana WM et la., 2003, Mannelli, L.D.C et l., 2009). However, other authors referred that there is no direct evidence (Ming-Feng Liao et al., 2022).
Several studies have demonstrated that apoptotic activities in injured dorsal root ganglia increased after injury in different rat models (Segushi M et al.,2009, Meyer L et al., 2009, Wiberg R et al. 2019), and some agents could simultaneously suppress pain behavior and apoptotic activities (Campana WM et la., 2003, Mannelli, L.D.C et l., 2009). However, there is no direct evidence that agents showing inhibition of apoptotic activities in the injured neurons could attenuate pain behavior (Ming-Feng Liao et al., 2022).
Comment 3. The parts “2. Pain – Definition and mechanisms – a brief review”, and “3.
Apoptosis and pain” are a little too long and need to be condensed. The authors may only keep the contents that will be discussed and mentioned in “4. Apoptosis and clinical implications”. “4.1. Apoptosis and neuropathic pain” can be merged into “4.3. Apoptosis in Pain therapy”.
Answer: We made some adjustments to the text. However, as this is a review article related to apoptosis and pain and as none of the other reviewers requested this change, we tried not to change the text too much.
Comment 4: In the description of “3. Apoptosis in Pain therapy…Isoflurane, a general inhalation anesthetic”, it seems a little weird that the authors explain the analgesic effects of isoflurane are through inhibiting apoptosis of myocardial cells. Is there evidence that isoflurane could inhibit apoptosis of neurons?
Answer: We add some text and references showing that isoflurane may reduce neuronal apoptosis (Masahiko Kawaguchi et al., 2004). However, these effects may be dose-dependent (Zhipeng Xu et al., 2011). We have also included other references showing the effects of other potential therapies. In part 4.2 we added the following text:
Further, Masahiko Kawaguchi et al., (2004) show that isoflurane reduced early the development of neuronal apoptosis in rats subjected to focal cerebral ischemia. Other studies demonstrated that isoflurane at low concentration attenuated the Aβ-induced reduction in Bcl-2/Bax ratio and leads to a mild elevation of cytosolic calcium levels. However, these results are dose-dependent, suggesting that isoflurane may have dual effects on Aβ-induced toxicity (protection or promotion) (Zhipeng Xu et al., 2011).
There are also scientific evidence showing that antioxidants may be used preventively and therapeutically to reduce not only oxidative stress related parameters but also inflammatory response and pain in several diseases (Carrasco et al., 2013, 2014a,b, 2018) and this effects could contribute to reduce apoptosis observed in neuropathic pain.
A randomized controlled study reported that L-carnosine exerted neuroprotective activity by significantly decreasing proinflammatory (NF-κB, TNF-α) and apoptotic (caspase-3) markers and increasing Nrf2 in colorectal cancer patients with oxaliplatin-induced PN [107]. Therefore, the use of Nrf2 inducers in treating neuropathic pain under clinical settings holds great promise in the future (Paramita Basu et al., 2022).
Comment 5: Figures 3 and 4 described the mechanisms of two different individual drugs. The author may make a single figure which presents all apoptotic pathways and summarize all drugs involved in apoptosis and pain in this single figure.
Answer: We have made a single figure which brings together figures 3 and 4, as suggested. We try to include drugs involved in apoptosis, some of which are in the text and table 1. However, in order not to get too full, we chose not to put everything since a part of the drugs/compounds are in table 1
Comment 6: In “table 1. Some preclinical studies in pain with biomarkers linked with apoptosis”, the authors should cite the reference to the individual drugs. Some drugs in this table are not involved in the apoptotic pathway (ex: rapamycin is a medication that can promote autophagy). The authors may create a new table that listed all drugs which were discussed in apoptosis and pain formation in this review.
Answer: As suggested, we have created a new table 1 (and changed the title), including only drugs/compounds involved in apoptosis that could be used to treat pain
Comment 7: The references style is not the same as the style suggested by biomedicines.
Answer: We corrected the reference style according to Biomedicines
Round 2
Reviewer 3 Report
In this revision, the authors added descriptions of the roles of TRP channels and oxidative stress in the apoptosis in chronic pain formation, and the effects of caspase inhibitors in neuropathic pain if different animal models. Grossly speaking, I believed the authors tried to have a detailed and comprehensive review. The contents of the article are also enriched after revision. However, the structure and order of this article are a little disorganized and some parts are not necessary.
There are some suggestions before publication.
- As I understand, the 2nd part is the introduction to the basic mechanisms of pain. However, I think some contents are the basic knowledge about this field and could be condensed. For example, the parts of the six keynotes of pain definition on page 3 are a little like plagiarism, and not relevant to the apoptosis in pain, and could be deleted.
- The 2nd part of the article (page 3 to page 5) discussed the basic molecular mechanisms of pain formation. However, “there are several ways to categorize pain…….” on page 4 discussed the terminology of pain, which is not relevant to the basic molecular mechanisms of pain formation in this section. Those descriptions can be deleted or moved to the beginning of 2nd part of the article.
- The end of 2nd part of the article discussed the role of TRPV and mitochondria in pain formation, which are an important part of this article. I suggest that the authors may have more references in this part. The authors may move the descriptions on “4.1 apoptosis and neuropathic pain” on page 17 to this section.
- The goals of the descriptions of “3.1: apoptosis cell signaling pathway” are not clear and consistent. Did the authors only wish to introduce the knowledge of signal pathways, or try to correlate the apoptosis pathway to pain formation? For example, the authors only introduce the general pathway of apoptosis, including the extrinsic and intrinsic pathways, without mentioning their relations to chronic pain formation in the part of “the extrinsic and intrinsic pathway”. However, in the parts of “the p38-MAPK pathway, ubiquitin-proteasome pathway, and oxidative stress”, the authors emphasized the relations between those pathways to pain formation, in addition to introducing the basic knowledge of the p38-MAPK and, ubiquitin-proteasome pathway.
- The part of “pyroptosis…” on page 5, “FADD/FAS/TRADD…” on page7, and “NGF, BDNF…” on page 8, are not relevant to the discussions of apoptosis and pain formation in the lateral parts of this article. The descriptions of those parts could be shortened or deleted.
- The part about “almost 20% of the European population …” at the end of page 6 is not relevant to the discussions about the basic mechanisms of the apoptosis pathway in the section. Those descriptions could be moved to the introduction or the beginning of 2nd part “pain-definition and mechanisms”.
- The subtitles of “3. Apoptosis and pain” are disorganized. For example, “3.2 Ubiquitin proteasome pathway and apoptosis” and “3.2 Oxidative stress” had the same subtitle “3.2”.
- In the “4. Apoptpsis and clinical implication”, it seems that the authors wish to discuss the clinical application of the medications which can modulate pain formation through adjusting apoptosis. I suggested that the authors could focus on discussions about the medications which could modulate apoptosis and pain formation in this part. For example, as mentioned before, “4.1 apoptosis and neuropathic pain” discussed the basic mechanisms of apoptosis again. I suggested those descriptions could be moved to “3.1 apoptosis and signaling pathways”.
- In the “4.2 Biomarkers and circulating mediators”, the authors discussed the basic molecules of oxidative damage without discussing the medications that could modulate oxidative stress and apoptosis. I suggested those descriptions can be merged to “3.2 oxidative stress”.
- It is good that the authors summarized different medications that have the potential to modulate apoptosis and treat neuropathic pain. I suggested that the authors could add more information to this table (ex: which animal models, which tissues have the molecules changes, DRGs or SDH?)
Author Response
We thank you for taking the time to assess our manuscript and for the constructive comments. We have addressed all your comments. Please, see below the answers to your specific comments. In the revised manuscript, we highlight in yellow the improved text using the “Track Changes” function.
Comment 1. As I understand, the 2nd part is the introduction to the basic mechanisms of pain. However, I think some contents are the basic knowledge about this field and could be condensed. For example, the parts of the six keynotes of pain definition on page 3 are a little like plagiarism, and not relevant to the apoptosis in pain, and could be deleted.
Answer: We have condensed some text and eliminate the six notes of pain definition as suggested.
Comment 2. The 2nd part of the article (page 3 to page 5) discussed the basic molecular mechanisms of pain formation. However, “there are several ways to categorize pain…….” on page 4 discussed the terminology of pain, which is not relevant to the basic molecular mechanisms of pain formation in this section. Those descriptions can be deleted or moved to the beginning of 2nd part of the article
Answer: We have deleted some of those descriptions/explanations as suggested.
Comment 3. The end of 2nd part of the article discussed the role of TRPV and mitochondria in pain formation, which are an important part of this article. I suggest that the authors may have more references in this part. The authors may move the descriptions on “4.1 apoptosis and neuropathic pain” on page 17 to this section.
Answer: We made some adjustments to the text and passed a paragraph from page 16/17, referred to by the reviewer, to the 2nd section. We also have add more references (and text) as suggested, and the text is now:
Comment 4: The goals of the descriptions of “3.1: apoptosis cell signaling pathway” are not clear and consistent. Did the authors only wish to introduce the knowledge of signal pathways, or try to correlate the apoptosis pathway to pain formation? For example, the authors only introduce the general pathway of apoptosis, including the extrinsic and intrinsic pathways, without mentioning their relations to chronic pain formation in the part of “the extrinsic and intrinsic pathway”. However, in the parts of “the p38-MAPK pathway, ubiquitin-proteasome pathway, and oxidative stress”, the authors emphasized the relations between those pathways to pain formation, in addition to introducing the basic knowledge of the p38-MAPK and, ubiquitin-proteasome pathway.
Answer: We want to introduce the knowledge of apoptosis signaling pathways, or try to correlate the apoptosis pathways to pain formation. We have deleted some text and have try to clarify the relation of this pathways (intrinsic and extrinsic) with pain.
Comment 5: The part of “pyroptosis…” on page 5,
- “FADD/FAS/TRADD…” on page7,
- and “NGF, BDNF…” on page 8,
are not relevant to the discussions of apoptosis and pain formation in the lateral parts of this article. The descriptions of those parts could be shortened or deleted
Answer: We have deleted all the suggested parts. We have only maintained the NGF, as this NT through its low-affinity p75 neurotrophin receptor (NGF-R/p75-NTR) activates caspase 8, as mentioned in figure 2 and now we hope that all these are clarified in the text.
Comment 6: The part about “almost 20% of the European population …” at the end of page 6 is not relevant to the discussions about the basic mechanisms of the apoptosis pathway in the section. Those descriptions could be moved to the introduction or the beginning of 2nd part “pain-definition and mechanisms”
Answer: We have move to the introduction the text suggested by the reviewer.
Comment 7: The subtitles of “3. Apoptosis and pain” are disorganized. For example, “3.2 Ubiquitin proteasome pathway and apoptosis” and “3.2 Oxidative stress” had the same subtitle “3.2”.
Answer: We have corrected the subtitles
Comment 8: In the “4. Apoptpsis and clinical implication”, it seems that the authors wish to discuss the clinical application of the medications which can modulate pain formation through adjusting apoptosis. I suggested that the authors could focus on discussions about the medications which could modulate apoptosis and pain formation in this part. For example, as mentioned before, “4.1 apoptosis and neuropathic pain” discussed the basic mechanisms of apoptosis again. I suggested those descriptions could be moved to “3.1 apoptosis and signaling pathways”
Answer: We have adjust the text and moved part of the text as suggested.
Comment 9: In the “4.2 Biomarkers and circulating mediators”, the authors discussed the basic molecules of oxidative damage without discussing the medications that could modulate oxidative stress and apoptosis. I suggested those descriptions can be merged to “3.2 oxidative stress”.
Answer: In this part (4.2) we described some potential biomarkers related with apoptosis and pain mediated not only by oxidative stress, but also inflammatory mediators and some proteasome constituents. We have made some adjustments in the text in order to clarify better what we want to say in this part and we have changed de subtile 4.2 to “Mechanisms related to nerve damage as pain biomarkers and/or circulating mediators”. In the next part (4.3) is that were we discuss the medications that could modulate apoptosis mainly through oxidative stress, but also other mechanisms.
Comment 10: It is good that the authors summarized different medications that have the potential to modulate apoptosis and treat neuropathic pain. I suggested that the authors could add more information to this table (ex: which animal models, which tissues have the molecules changes, DRGs or SDH?)
Answer: We have already described in the text and/or in the figure 3 and summarized in table 1 the different potential medications that have the potential to modulate apoptosis and treat neuropathic pain. As suggested by the reviewer we add to the table a new colunn with the model used to test this new drugs (as the mechanism was already in the table),

Round 3
Reviewer 3 Report
In this revision, the authors changed the order of the contents and condensed some descriptions. The quality of the manuscript had been improved and became more easily to be read.
There are some minor suggestions before publication.
- Grossly speaking, the authors discussed three major topics of apoptosis in pain formation: (TRP) channels, ROS, and proinflammatory cytokines. There was only a smaller part that discussed the p38 in the apoptosis and neuropathic pain. So, the description “The degree of dorsal root ganglion (DRG) neuronal apoptosis has recently been associated with spinal nerve injury via the caspase signaling and/or Protein Kinase A (PKA) activation through the p38 MAPK pathway, generating and maintaining neuropathic pain.” in the abstract could be deleted. Besides, transient receptor potential (TRP) cation channels can be added as a keyword.
- The authors seem not to have further descriptions of the “six main explanatory notes” in the manuscript. So the descriptions of “It has been expanded by the addition of six main explanatory notes that help to contextualize and better understand the concepts” in part 2 could be deleted.
- Figure 1 did not have a figure legend.
- As mentioned before, the authors discussed three major topics of apoptosis in pain formation: (TRP) channels, ROS, and proinflammatory cytokines. The authors could add some descriptions of the role of proinflammatory cytokines in apoptosis and pain formation at the end of part 2 “Pain – Definition and mechanisms – a brief review”.
- The title of “3. Apoptosis and pain” could be considered a change to “3. The mediator in the apoptosis pathway in pain formation.”
- I suggested adding a section about transient receptor potential (TRP) cation channels in the part of “3. Apoptosis and pain”. The descriptions of TRP in pain formation on page 10, page 14, and page 18 could be summarized in this independent section.
- On page 10 “The p38MAPK pathway” did not have a subtitle number (3.2 or 3.3?). As mentioned before, there were only a few descriptions of p38 in apoptosis and neuropathy pain formation. The descriptions of “The p38MAPK pathway” can be placed in the last part of part 3.
- “4.1 apoptosis and neuropathic pain” could be deleted, and the contents could be moved to other parts of the manuscripts as in previous suggestions (ex: to a new section of TRP in pain formation).
- “4.2. Mechanisms related to nerve damage as pain: biomarkers and/or circulating mediators” could be changed to “Biomarkers and circulating mediators in pain”.
- “4.3 Apoptosis in Pain therapy” could be changed to “4.3 Analgesic agents through modulating apoptosis activities”.
- In table 1, it is confusing that the column of “Model” included the experimental model (ex: rats with spinal cord injury, rats with chronic constriction injury) and the localization (ex: DRG neurons, spinal dorsal horn). The authors could add another column to describe the localization of molecular changes. Did the “6 weeks, 4 weeks” in the frequency of administration” mean “every 6 weeks, every 4 weeks”?
Author Response
We thank you for taking the time to assess our manuscript and for the constructive comments. We have addressed all your comments. Please, see below the answers to your specific comments. In the revised manuscript, we highlight in yellow the improved text using the “Track Changes” function.
Comment 1. Grossly speaking, the authors discussed three major topics of apoptosis in pain formation: (TRP) channels, ROS, and proinflammatory cytokines. There was only a smaller part that discussed the p38 in the apoptosis and neuropathic pain. So, the description “The degree of dorsal root ganglion (DRG) neuronal apoptosis has recently been associated with spinal nerve injury via the caspase signaling and/or Protein Kinase A (PKA) activation through the p38 MAPK pathway, generating and maintaining neuropathic pain.” in the abstract could be deleted. Besides, transient receptor potential (TRP) cation channels can be added as a keyword.
Answer: We have deleted from the abstract the text sugested by the reviwer and added the transient receptor potential (TRP) cation channels as a keyword.
Comment 2. The authors seem not to have further descriptions of the “six main explanatory notes” in the manuscript. So the descriptions of “It has been expanded by the addition of six main explanatory notes that help to contextualize and better understand the concepts” in part 2 could be deleted.
Answer: We have deleted the descriptions/explanations as suggested.
Comment 3. Figure 1 did not have a figure legend. ?? The text that was sent to us the figure1 has the corresponding title
Answer: We have add a legend to the figure 1 as suggested. “The figure shows the interactions between distinct parts of a nociceptor (periphery, dorsal root ganglion and spinal cord) with different types of non-neurons cells (keratinocytes, Schwann cells, satellite glial cells, oligodendrocytes and astrocytes), immune cells (macrophages, T cells and microglia), cancer cells, and bone marrow stem cells. These cells produce pro-nociceptive (highlighted in blue) and antinociceptive (highlighted in red) mediators, that modulate the nociceptor sensitivity and excitability through binding to their respective receptors. In the spinal cord dorsal horn , the central terminal of the nociceptor forms a nociceptive synapse with a postsynaptic neuron, to mediate pain transmission in the CNS”.
Comment 4: As mentioned before, the authors discussed three major topics of apoptosis in pain formation: (TRP) channels, ROS, and proinflammatory cytokines. The authors could add some descriptions of the role of proinflammatory cytokines in apoptosis and pain formation at the end of part 2 “Pain – Definition and mechanisms – a brief review”.
Answer: A paragraph concerning the role of proinflammatory cytokines was added (with references) at the end of part 2 of the manuscript as asked.
“It should also be noted that apoptosis is also regulated by proinflammatory cytokines, such as IFN-gama, TNF-alfa,IL-1beta, and IL-6 (10.3390/ijms23052685). The role of IL-1 and TNF-alfa in apoptosis in neurodegenerative diseases such as Alzheimer’s disease was previously described (doi: 10.1038/35037739; 10.3390/ijms23052685). More importantly, it is known that these proinflammatory cytokines have crucial roles in the modulation of neuropathic pain (10.3390/ijms23052685; doi: 10.1038/nri3621).”
Comment 5: The title of “3. Apoptosis and pain” could be considered a change to “3. The mediator in the apoptosis pathway in pain formation.”
Answer:
In our opinion, the title we have proposed is better suited to what is intended, but we have changed to “3. Apoptosis pathways as mediators of pain formation”. We hope that this new title is more in line with the reviewer's proposal.
Comment 6: I suggested adding a section about transient receptor potential (TRP) cation channels in the part of “3. Apoptosis and pain”. The descriptions of TRP in pain formation on page 10, page 14, and page 18 could be summarized in this independent section.
Answer: Histories can be told with different plots. This is our initial line of thought now added with the valuable input of the reviewers that enriched the manuscript. However, we do not agree with these particular changes proposed by the reviewer. Independently, the authors arrived at the conclusion that those suggestions alter radically the primary design of the manuscript. Therefore, we opted for maintaining the basic structure of the manuscript. However, if the reviewer insists on the changes we can alter the sections as suggested.
Comment 7: On page 10 “The p38MAPK pathway” did not have a subtitle number (3.2 or 3.3?). As mentioned before, there were only a few descriptions of p38 in apoptosis and neuropathy pain formation. The descriptions of “The p38MAPK pathway” can be placed in the last part of part 3.
Answer: We have add a number to the subtitle “The p38MAPK pathway” and placed in the last part of part 3, being the number 3.6., as suggested by the reviewer. We also add references showing the relevance of P38MAPK pathway in pain (doi.org/10.1016/j.bbadis.2009.12.009, DOI:10.2217/nnm-2018-0054; doi: 10.3390/ijms19030867) and add a paragraph confirming its relevance “Further, it has been suggested that the inhibition of either JNK or p38 may represent a potent clinical target for neuropathic pain management (doi: 10.3390/ijms19030867)”.
Comment 8: 4.1 apoptosis and neuropathic pain” could be deleted, and the contents could be moved to other parts of the manuscripts as in previous suggestions (ex: to a new section of TRP in pain formation.
Answer: Histories can be told with different plots. This is our initial line of thought now added with the valuable input of the reviewers that enriched the manuscript. However, we do not agree with these particular changes proposed by the reviewer. Independently, the authors arrived at the conclusion that those suggestions alter radically the primary design of the manuscript. Therefore, we opted for maintaining the basic structure of the manuscript. However, if the reviewer insists on the changes we can alter the sections as suggested.
Comment 9: 4.2. Mechanisms related to nerve damage as pain: biomarkers and/or circulating mediators” could be changed to Biomarkers and circulating mediators in pain”
Answer: We have changed the title of 4.2 as suggested by the reviewer. 4.2. “Biomarkers and circulating mediators in pain”
Comment 10: 4.3 Apoptosis in Pain therapy” could be changed to “4.3 Analgesic agents through modulating apoptosis activities”.
Answer: We agree that the title could be changed, but we considered that the title “Pain therapy through modulating apoptosis activities”, is more adequated with the described in this section, including the table 1, as some of the refered compounds/approaches are not yet included in the class of analgesic agents (besides its analgesic effect). We hope that the reviewer agree with this little modification relative to the suggested.
Comment 11: In table 1, it is confusing that the column of “Model” included the experimental model (ex: rats with spinal cord injury, rats with chronic constriction injury) and the localization (ex: DRG neurons, spinal dorsal horn). The authors could add another column to describe the localization of molecular changes.
Answer: Table 1 was improved with more information added concerning the “Administration and time of the experiment”, “Model” and “Localization of changes”. We hope that the information and mode of presentation are now clearer.
Comment 12:. Did the “6 weeks, 4 weeks” in the frequency of administration” mean “every 6 weeks, every 4 weeks”?
Answer: With the new version of the table, there are no doubts concerning the frequency of administration.
